# Urban rats are the 'fall-guy': Resident motivations for municipal rat complaints

**Michael Joseph Lee** [1,2]*, **Kaylee A. Byers**[1,3], **Xiaocong Guo**[1,2], **Lisa K. F. Lee**[1,4], **Susan M. Cox**[2,5], **Chelsea G. Himsworth**[1,2,6]

**1** Canadian Wildlife Health Cooperative, Animal Health Centre, Abbotsford, BC, Canada, **2** School of Population and Public Health, University of British Columbia Canada, Vancouver, BC, Canada, **3** Pacific Institute on Pathogens, Pandemics and Society, Simon Fraser University, Burnaby, BC, Canada, **4** Department of Veterinary Pathology, Western College of Veterinary Medicine, University of Saskatchewan, Saskatoon, SK, Canada, **5** The W. Maurice Young Centre for Applied Ethics, University of British Columbia, Vancouver, BC, Canada, **6** Animal Health Centre, British Columbia Ministry of Agriculture, Abbotsford, BC, Canada

* m.lee@ubc.ca

**Data Availability Statement:** Data used in this study was obtained from the City of Vancouver and is governed by the Freedom of Information and Protection of Privacy Act. The findings and conclusions in this publication do not reflect the

## Abstract

Rats are an important issue in cities globally. Despite their ubiquity, perceptions and concerns about rats vary with circumstance and the context in which a person interacts with them. Municipal rat management programs are a service to communities and therefore must be responsive to the varied concerns of their residents. Understanding why communities are concerned about rats can help inform rat management programs to meet the specific needs of their residents. The objective of this study was to identify why the residents of Vancouver, Canada care about rats and what they want done to address them. To do this, we qualitatively analyzed 6,158 resident complaints about rats made to the city's municipal government between January 2014 and May 2020. Using a qualitative descriptive coding process, we found that rats were a priority in a minority of cases. In general, people were more concerned about broader community issues, such as neighborhood disorder, of which rats were one part. Complaints tended to be made when problems were highly visible, nearby, and when the complainant wanted the city to take action to alleviate this issue, particularly when they were in and around their living spaces. The rates of complaints were highest in the most economically and socially deprived neighborhoods and lowest in the most privileged neighbourhoods. We synthesize this information with a view towards understanding how to develop objectives and actions for municipal management strategies that are grounded in community concerns.

## 1. Introduction

Rats, in the genus *Rattus*, are present in cities around the world where they have important public health and economic impacts. These animals can carry many pathogens transmissible to humans [1], they increase the levels of indoor allergens associated with heightened asthma morbidity [2,3], and they negatively impact people's mental health by causing stress and

opinions of the data stewards. Data stewards provided this data to the study authors specifically for the purposes of informing the development of suitable rat management protocols for Vancouver and cannot be shared by study authors. Data is held by the City of Vancouver and readers can refer to https://vancouver.ca/your-government/freedom-of-information.aspx to request access to data or they can contact the Access to Information and Privacy office at foi@vancouver.ca.

**Funding:** This work was funded by the City of Vancouver's 2018-2020 Street Cleaning Grant Program in support of their efforts to establish suitable rat management protocols for Vancouver. The funders had no role in study design, data collection and analysis, decision to publish, or preparation of the manuscript.

**Competing interests:** The authors have declared that no competing interests exist.

anxiety [4,5]. Rats can also cause economic strain, by damaging and degrading roads, wires, buildings, other urban infrastructure [6] and by destroying personal belongings and food products [7,8]. Furthermore, they can directly impact local businesses through the costs associated with rodent control, and their presence leads to business closures and damaged reputations [9].

Despite these widespread impacts, people's perceptions of and concerns about rats may vary geographically. For example, residents of Niamey, Niger reported being most concerned about rats damaging their food stocks, homes, and other personal belongings [8]. In comparison, residents in Chicago, USA recounted apprehension related to personal safety and disease spread [5], while interviewees living in an impoverished neighborhood in Vancouver, Canada reported that rats disrupted sleep and elicited feelings of anxiety [10]. Peoples' concerns related to rats may vary because they are entwined with a variety of municipal issues. For instance, in the Canadian study, rats were symbolic of social neglect and disregard for neighborhoods [10]. To interviewees, rats were interconnected with other issues affecting their neighbourhoods including homelessness and the ongoing opioid crisis [11]. The interviewees prioritized these other issues above rat management. While there are few studies on human perceptions of rats, they collectively suggest that the perceived priority of rat problems is likely to vary across space, time, and with other factors such as socio-demographic characteristics.

Understanding why people are concerned about rats is important for the development of effective rat management strategies. This is because municipal rat management programs are designed to be a service to their communities and therefore they must be responsive to resident concerns to effectively meet their needs [12,13]. However, there is limited published information available on the specific reasons that urban residents care about rats and where, when, and what they want municipal governments to do about them. In addition, it is important to define these perceptions locally because residents' needs regarding rats are likely to vary between different cities and over time.

The objective of this study was to examine and describe rat complaints made by residents of Vancouver, Canada to the municipal government. To do this, we qualitatively analyzed resident complaints about rats made to the city's municipal government and assessed their spatial distribution. Specifically, we sought to: 1) infer whether rats were the primary reason a resident complained to the city or whether they were secondary to another issue; 2) describe the concerns that residents cited as their reason for complaining about rats; 3) identify the actions that residents requested of the city to address the issue; and 4) assess whether rat complaints were associated with neighborhood deprivation. We synthesize this information with a view towards understanding how to develop objectives and actions for municipal rat management strategies that are grounded in community concerns.

## 2. Material and methods

### 2.1 Municipal rat complaints data

Data analyzed in this study consisted of complaints about rats made to the municipal government of the City of Vancouver. Complaints were made by residents via the online VanConnect platform or via the 311-phone line [14]. Online submissions were written by the complainant. Phone calls were summarized by a phone operator. Each complaint included: (1) a complaint written by the phone operator or by the complainant; (2), the forward sortation area location or FSA (i.e., the first 3 digits of the 6-digit zip code) of the complaint; (3) the date of the complaint, and; (4) the date the case was closed by the City. The data was pre-categorized by the City of Vancouver into 29 categories (S1 Table) based on their content (i.e., property use complaint, citizen feedback case, dead animal pick-up, etc.).

Between January 2014 and May 2020, the city of Vancouver received 6,158 complaints that included the keywords "rat", "rats", "rodent", and "rodents". Data were accessed by study authors on 20 May 2020. No identifying information was provided. Entries were initially excluded if they were either about mice or were otherwise unrelated to rats, such as those regarding the 'rat race' (n = 604). Entries were also removed if they were pet rat-related or were requests for a dead animal pick-up (n = 1,269). Dead animal pickups involved cases where a complainant requested that the city remove a dead rat but contained no additional information. The final dataset consisted of 4,285 complaints. This study was approved by the University of British Columbia's human research ethics board (H19-02715).

## 2.2 Deprivation data

To examine the association between the number of complaints and neighborhood socio-economic status, we downloaded the Material and Social Deprivation Indices (MSDI). The MSDI, developed by the Institut National de Santé Publique du Québec [15], was created from six indicators in the 2016 Canadian census and the 2011 National Household Survey. The MSDI is comprised of two indices: (1) the material deprivation index, a measure of access to material resources and was developed using indicators such as employment status, income, and education and (2) the social deprivation index, a measure of the strength of social networks and was developed using indicators such as the number of single parent families, the proportion of people living alone, and martial status. In this study, we created a combined index of material and social deprivation using the first strategy recommended in the MSDI user guide [16]. The combined index has five categories including: (1) materially and socially privileged; (2) average material and social deprivation; (3) materially privileged but socially deprived; (4) materially deprived but socially privileged; and (5) materially and socially deprived. We linked complaints to neighborhood deprivation by the FSA and graphically examined whether the rate of complaints was associated with deprivation.

## 2.3 Qualitative data analysis

The overarching goal of this analysis was to understand why Vancouver residents were concerned about rats and what actions they wanted taken as a result of their complaint. To achieve this goal, we applied a method called qualitative description [17] in which we focused on describing and summarising the content of the textual complaints data using a qualitative coding process. Overall, this coding process (Table 1) involved defining categories (i.e., themes) to describe the content of the complaints, assigning each complaint to those categories, and then summarising the content of the complaints in each category. 'Qualitative codes' refer to the words or phrases used to label each category.

To begin this qualitative coding process (Table 1), we derived a preliminary group of descriptive qualitative codes based on 20% of all complaints. To do this, four coders were each randomly assigned 5% of all complaints, stratified by the 29 categories predefined by the City of Vancouver. Each coder independently developed a visual map of qualitative concepts [18] that described why the complainant cared about rats (e.g., rats were spreading garbage) and what they wanted to occur because of their complaint (e.g., rat extermination). This concept map was then used to develop a list of qualitative codes (i.e., 'garbage' and 'extermination') to categorize the content of the complaints. For example, if a complainant was calling to report that rats were spreading garbage in alleyways and was requesting that the city exterminate the rats, the call was coded as: 'resident concern about rats: spreading garbage' and 'action requested: extermination'. Coders collaboratively built a single set of qualitative codes to be applied across all complaints (Fig 1).

**Table 1. Details of the data analysis.**

| Step | Description |
|---|---|
| **1) Initial concept map and coding** | a. Four coders (MJL, KAB, LL, XG) were each assigned a random selection of 5% of the data*<br>b. Four coders independently developed:<br>    i. Visual concept maps to describe the content of their subset<br>    ii. Qualitative codes describing the content of the calls related to why the complainant was concerned about rats and what they wanted to occur because of their complaint<br>c. Coders compared and contrasted maps and codes<br>d. Coders collaboratively built a single concept map and set of codes for the entire 20% random selection of data |
| **2) Coder calibration** | a. The four coders independently mapped and coded the same 50 complaints<br>b. Compared maps and codes<br>c. Discussed discrepancies and emergent information<br>d. Came to a consensus on each discrepancy before proceeding |
| **3) Coding and concept mapping** | a. Each of the four coders was randomly assigned 25% of the overall dataset<br>b. Each coder independently:<br>    i. Qualitatively coded their subset of the data<br>        • Categorized each complaint by the co-developed qualitative codes<br>        • The initial set of codes was not changed throughout the coding process to facilitate consistency among coders and enumeration across the entire dataset<br>    ii. Refined the initial collaborative concept map to account for new information not captured by the qualitative codes developed in step 1<br>        • Individual concept maps were built through the addition of new concepts, restructuring, and reorganization as new information emerged from their respective datasets<br>        • Memoing was used to track reasoning behind changes<br>        • Maps were continually updated until no new call triggered the addition of new information to the map and the content of every call could be categorized within the framework |
| **4) Collaborative concept mapping** | a. Coders compared and contrasted concept maps<br>b. Discussed and agreed upon discrepancies in maps<br>c. Organized concepts into higher level themes<br>d. Collaboratively built a visual thematic framework to describe the content of the overall dataset |
| **5) Coding summaries** | a. Descriptive statistics were used to assess the proportion of each code in the overall dataset<br>b. Summaries were written for each code and are described in the Results section |

*All random selection occurred within categories of complaints predefined by the city. The city categorized calls into 29 categories (i.e., property use complaint, fire safety hazard, etc), such that for step 1 in this table, 5% of the codes were randomly selected from each of these 29 categories to give each coder 5% of the overall number of complaints.

For the main analysis, each coder was randomly assigned a non-overlapping 25% of the entire dataset to which they applied a two-step coding framework. First, each coder systematically categorized each complaint by the initial set of qualitative codes (Fig 1). Second, to capture new or emergent information that was not accounted for by the initial set of qualitative codes, each coder independently defined new codes as they encountered content that was not encompassed by the initial set of qualitative codes.

Finally, we tallied the total number of complaints that were categorized into each code in the initial qualitative coding framework (Fig 1). We did not enumerate emergent codes that were defined to capture new information that arose during the coding process because this was done independently by each coder. Instead we present this information descriptively to provide additional context to the results.

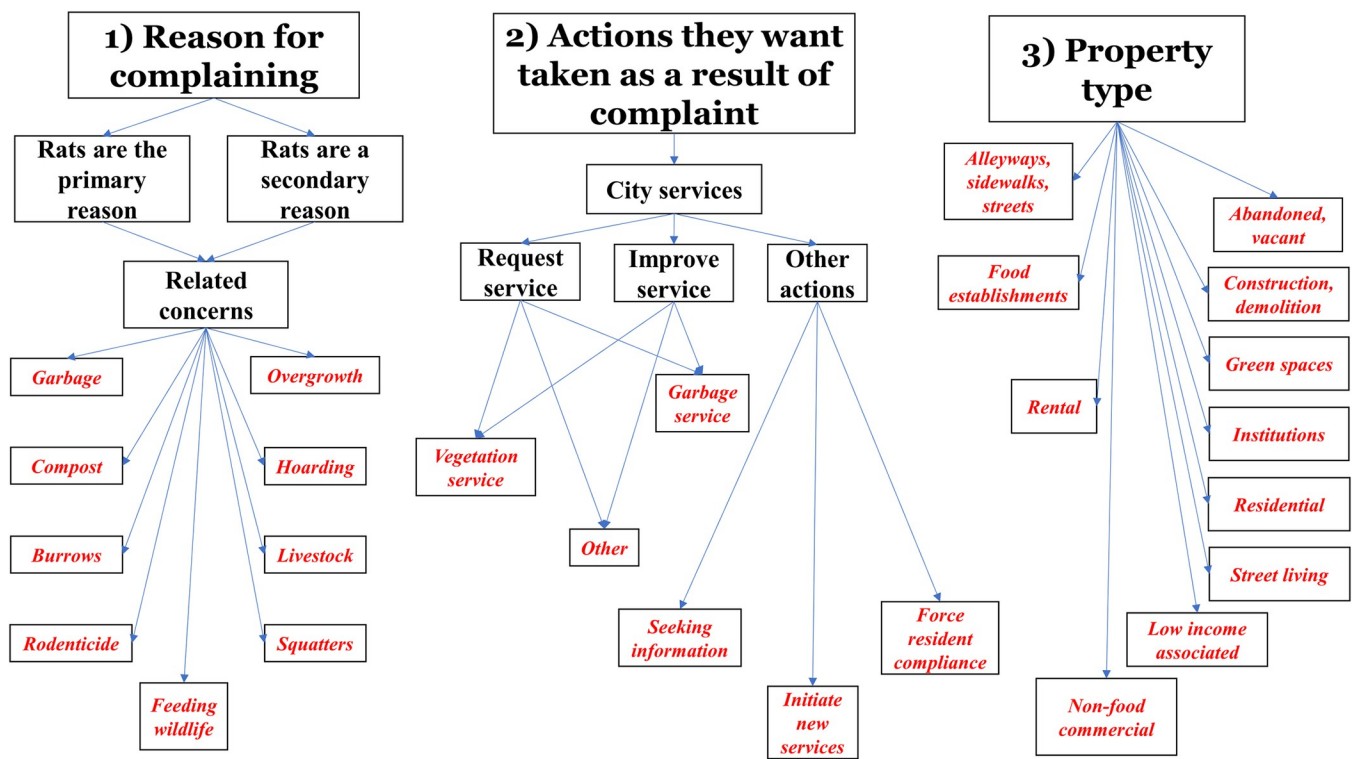

**Fig 1. Qualitative codes for enumeration.** Codes were categorized into three overarching groups: (1) reasons for complaining, (2) actions they want taken as a result of their complaint, and (3) property type the complaint was about. Red, italic font indicates codes that were used for qualitative coding and were enumerated.

## 3. Results

### 3.1 General description of complaint content

Complaints ranged from between one to twenty sentences in length and varied in detail. The most detailed complaints included information about why the individual was concerned about rats (e.g., health concerns), where the problem was (e.g., alleys), what other issues were associated with rats (e.g., waste), who they viewed as responsible for the problem (e.g., neighbours), how they wanted the city to address the issue (e.g., pest control services), as well as details about how the situation made them feel and whether or not they had complained about this issue previously. The least detailed complaints provided only a few words ('messy yard attracting rats'). As a result, some complaints were coded with all codes (i.e., reason for calling, action requested, property type, etc.) while others were only coded with one (i.e., action requested). When a particular code was not a part of the complaint, that complaint was not categorized for that code. Because of this, percentages are reported relative to the total number of complaints (n = 4,285) and percentages do not add up to 100% in each category.

### 3.2 Why were people complaining about rats?

Rats were the primary reason for the complaint in one third (1283 of 4286 complaints: 29.9%) of complaints (Fig 2A). In these instances, residents were concerned about: (1) rats damaging their property (e.g., chewing on car wires, damaging commercial goods); (2) health risks (e.g., diseases, asthma, breathing problems); (3) vulnerability of elderly people (e.g., parents were elderly and unable to address the issue themselves); (4) vulnerability of children (e.g., rats near

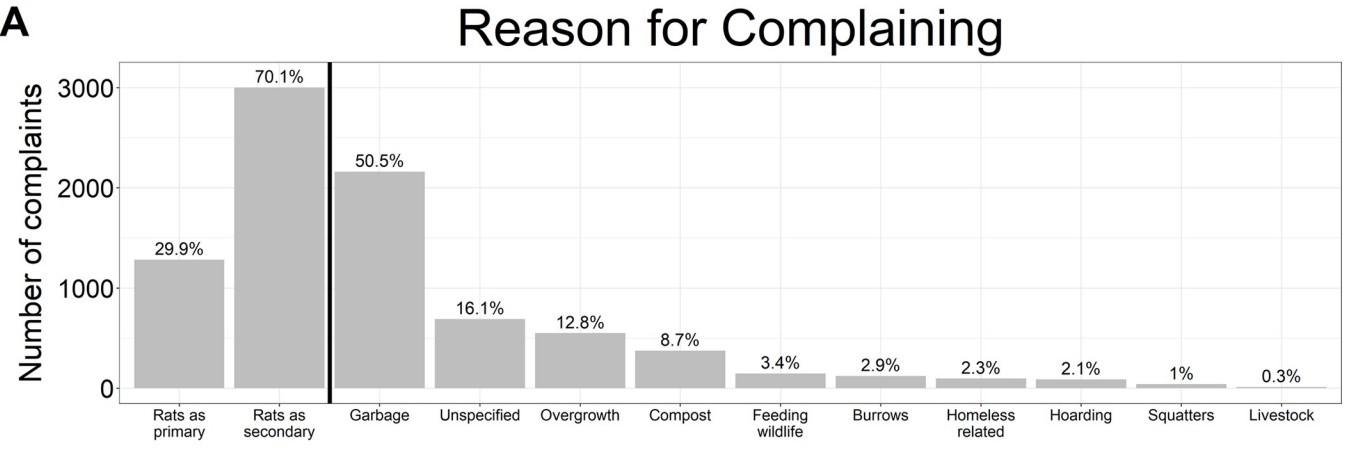

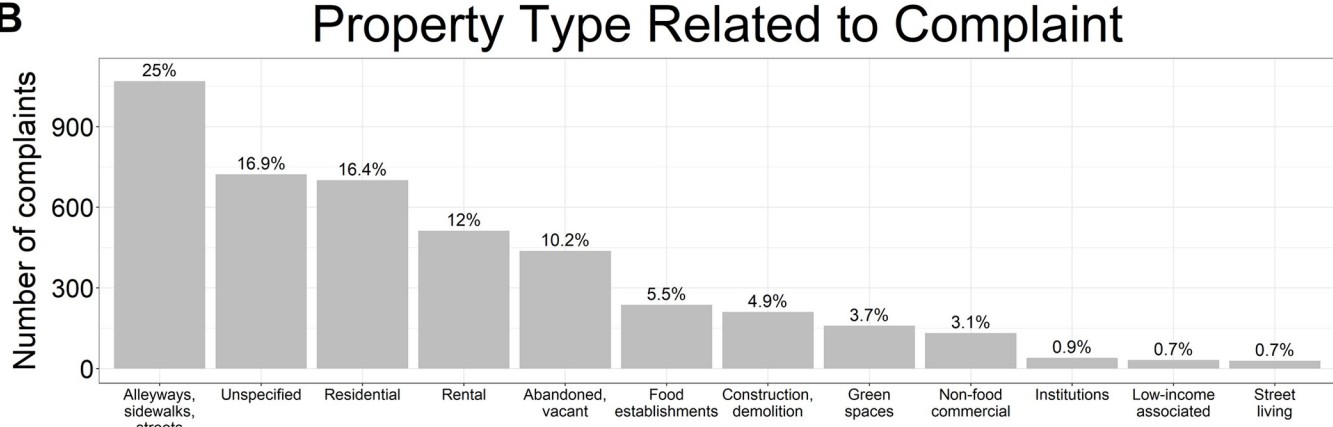

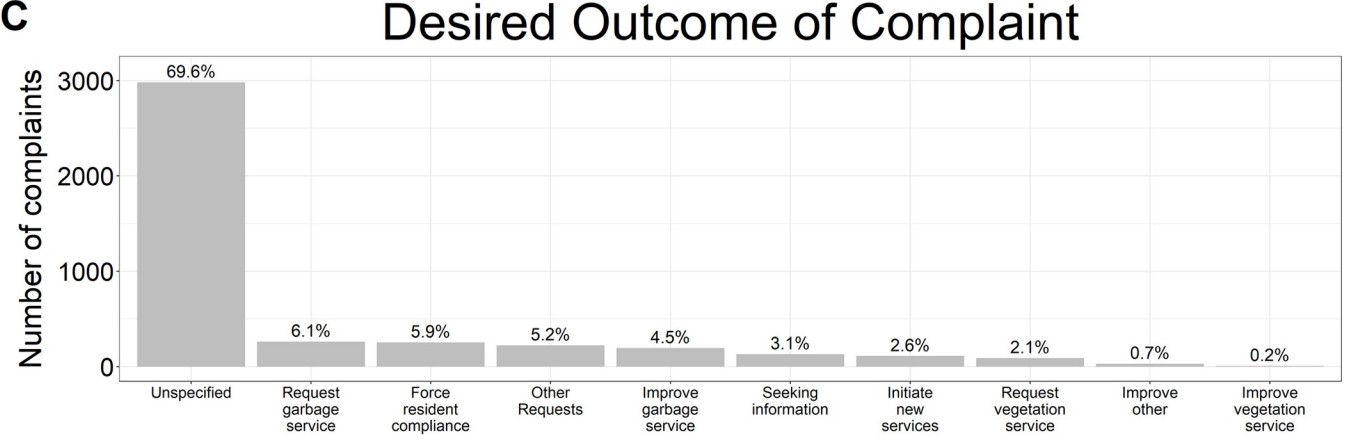

**Fig 2. Reasons that people complained about rats in the City of Vancouver.** Panel (A) shows the primary reason for complaining; panel (B) shows the property type the complaint was about; and panel (C) shows the actions that the complainants wanted the City to take as a result of their call.

playground/daycare); (5) the smells and sounds associated with rats (e.g., scratching in the wall); (6) impacts on sleep quality; and (7) rodenticide application, which was primarily considered a rat-associated health risk.

When rats were the primary concern, they were described as problematic in relation to their proximity, the potential for infestation of personal property, a neighbourhood infestation,

and in response to coverage of rats seen in the media. First, individuals were most concerned about rats when they were currently in or near their personal property (proximity). For example, people were concerned when rats were inside their home or building, in their yard, or in a neighbor's yard. Second, people were concerned when rats had the potential to infest their property (potential infestation). For instance, callers reported poorly maintained city-owned vegetation near their homes because they felt it might provide rats with a route into their building via branches overhanging the roof. Third, callers expressed concern when there were new rat sightings in a neighbourhood, a visible infestation in the neighbourhood that was perceived to be growing, or rats appeared to be moving into the neighbourhood from surrounding areas (e.g., community gardens or nearby properties (neighbourhood infestation). Finally, individuals complained about rats in response to hearing about them in the news (media coverage). In these cases, complainants had a heightened awareness of rats because of the news, or they disagreed with the report because they felt that the city did not have a rat problem.

The majority of complaints (3002 of 4285 complaints: 70.5%) identified rats as a secondary issue. In these instances, callers discussed rats as one symptom of another broader community issue that they were primarily concerned with, such as waste management and vacant houses. Where rats were the secondary concern, they were often presented as a part of the justification for why the primary issue they reported required immediate attention. For example, numerous complainants were upset with how neighbors were not managing their waste/garbage. The complainant would describe how neighbours' waste was either attracting rats or providing a place for rats to thrive.

Overall, 83.9% of complaints (3595 of 4285 complaints) described broader community issues that were perceived to be associated with rats while the other 16.1% of complaints did not specify a concern other than rats (Fig 2A). The framing of these broader community issues differed depending on whether an individual was reporting rats as a primary or secondary issue. Where rats were the primary concern, complainants cited these community issues as causing or contributing to a rat infestation. Where rats were the secondary concern, complainants considered these issues important in and of themselves, but with potential to promote rats. For example, many complainants reported neighbourhood garbage or compost issues that were unsightly, unsanitary, smelly, and which, if left unmanaged, might attract rats.

The most common issue associated with rats was garbage (2163 of 4285 complaints: 50.5%). These included reports of abandoned garbage, overflowing receptacles, incorrect garbage storage (e.g., in bags, dumpster left open or unlocked), broken trash bins, poor trash bin maintenance (e.g., dumpsters left unplugged), delayed or missed garbage pickup, and junk/debris/waste in yards or alleyways. Although these issues were often viewed as causes of infestations, people also felt that rats perpetuated and worsened these problems by getting into garbage and spreading it throughout the neighbourhood.

The second most common issue associated with rats was overgrowth of vegetation (550 of 4285 complaints: 12.8%). These complaints were generally about poorly maintained vegetation in a nearby yard or in the greenspaces between a building/home and the road or sidewalk. For complainants who were primarily concerned about rats, overgrowth was seen as harbourage for rats. For complainants secondarily concerned about rats, the overgrowth was described as unsightly and undesirable in the neighbourhood. Overgrowth and waste issues were often co-reported as an issue of overall property maintenance. For example, individuals reported neighbours who neglected yard maintenance, allowing for overgrowth and the accumulation of junk, debris, and fruit from unharvested trees on the ground.

The third most reported issue was compost (374 of 4285 complaints: 8.7%). Similar concerns regarding garbage were also applied to compost (i.e., overflowing, poorly maintained, or broken compost bins). For those who were primarily concerned about rats, compost was

described as a major food source for rats because: (1) rats were able to chew through the residential compost bins; (2) compost haulers dropped compost along the alleyways; (3) haulers damaged compost bins allowing for access by rats; and (4) some people did not store compost in closed containers. These complainants reported an intense dislike and/or disgust of composting in general. Rats were one of many groups of pests seen to be attracted by compost, including coyotes, bugs, birds, and raccoons.

Less commonly reported issues associated with rats included feeding wildlife (147 of 4285 complaints: 3.4%), rat burrows (123 of 4285 complaints: 2.9%), homelessness (97 of 4285 complaints: 2.3%), and hoarding (88 of 4285 complaints: 1%).

## 3.3 Locations of complaints

**3.3.1 Geographical distribution and deprivation.** In the study area there are 29 FSAs and the mean number of complaints per FSA (range) was 6.8 (0.6–14.7) per 1,000 population (Fig 3A). In general, the rate of complaints increased with increasing material and social deprivation (Fig 3A). However, this association was primarily driven by the most privileged and the most deprived FSAs, which contained the lowest and highest complaint rates, respectively. FSAs with intermediate levels of deprivation had a greater range of complaints (Fig 3B).

**3.3.2 Location related to complaint content.** Complaints were made in relation to several community locations. The most common areas were: (1) alleyways, sidewalks, and streets (1070 of 4285 complaints: 25%); (2) residential homes and properties (701 of 4285 complaints: 16.4%); (3) residential rental homes and properties (513 of 4285 complaints: 12%); and (4) abandoned or vacant properties (438 of 4285 complaints: 10.2%; Fig 2B). Complaints were often coded with multiple overlapping locations (e.g., "alleyways, sidewalks, and streets" in a "residential" area).

In most instances, complainants reported issues that were near their residence. For instance, people complained about alleyways and sidewalks near their home or that they visited frequently. Similarly, reports regarding residential and rental properties were typically made about properties they lived in, near, or frequently passed.

Some individuals used property types to justify why they were directing their complaint to the municipal government. For example, many complaints were made about garbage issues in neighbors' backyards. Some of these complaints justified the need to report because the garbage and associated rat issue was spilling over into city-owned property such as the alleyway– thus requiring city involvement. Others reported that the problem originated outside of their property or was on such a large scale that the municipal government was the only group with jurisdiction to manage it. Complainants also described rat problems originating on a single property as the source of issues for the whole neighborhood.

## 3.4 Actions requested

Most complaints (2982 of 4285 complaints: 69.6%) did not directly request a specific action. These complaints simply reported a rat issue or associated problem. For example, many complaints reported the presence of rats, garbage, and/or overgrowth in an area. It was not clear whether the complainant wanted the reported issue fixed by the city or whether they wanted to alert the municipality.

For those complaints that did include a request for action (Fig 2C), actions requested included: (1) picking up garbage (262 of 4285 complaints: 6.1%); (2) forcing residents to fix a problem (254 of 4285 complaints: 5.9%) such as clearing their yard of clutter; (3) improving existing garbage services (194 of 4285 complaints: 4.5%) including increasing the number of trash pickups, supplying better garbage bins, and changing the hauling company; (4) seeking

**A**

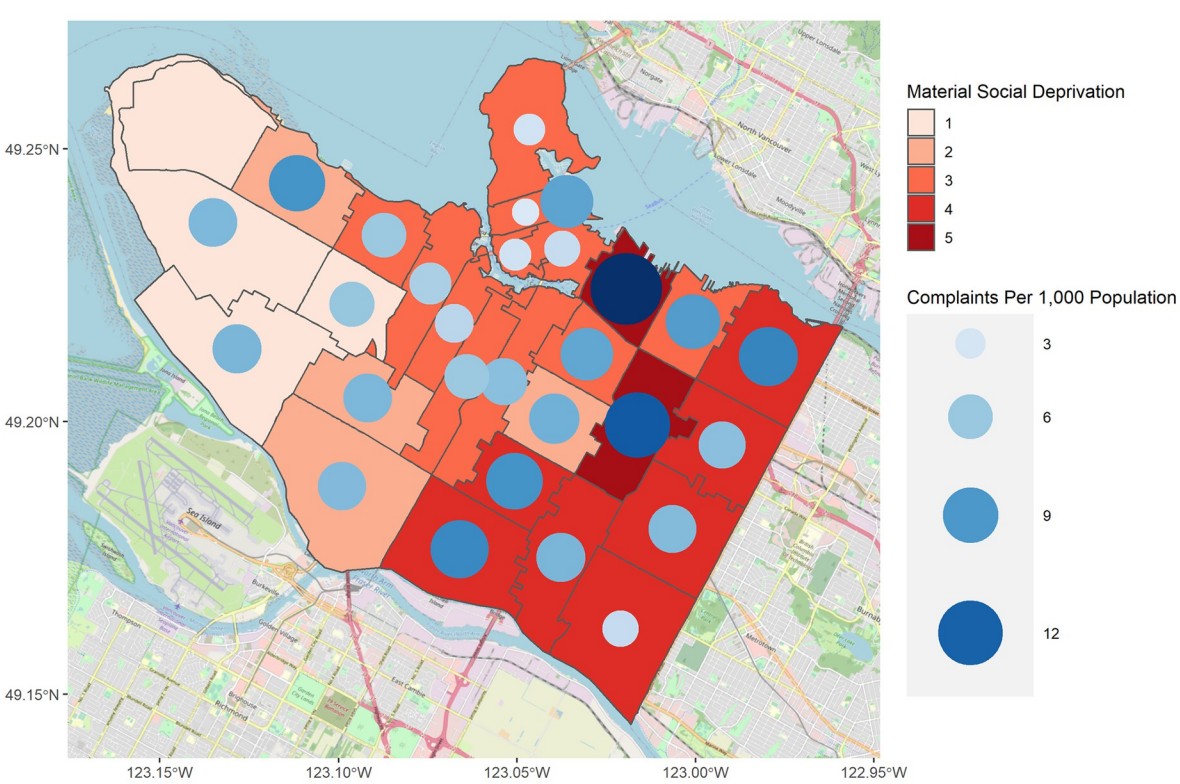

**B**

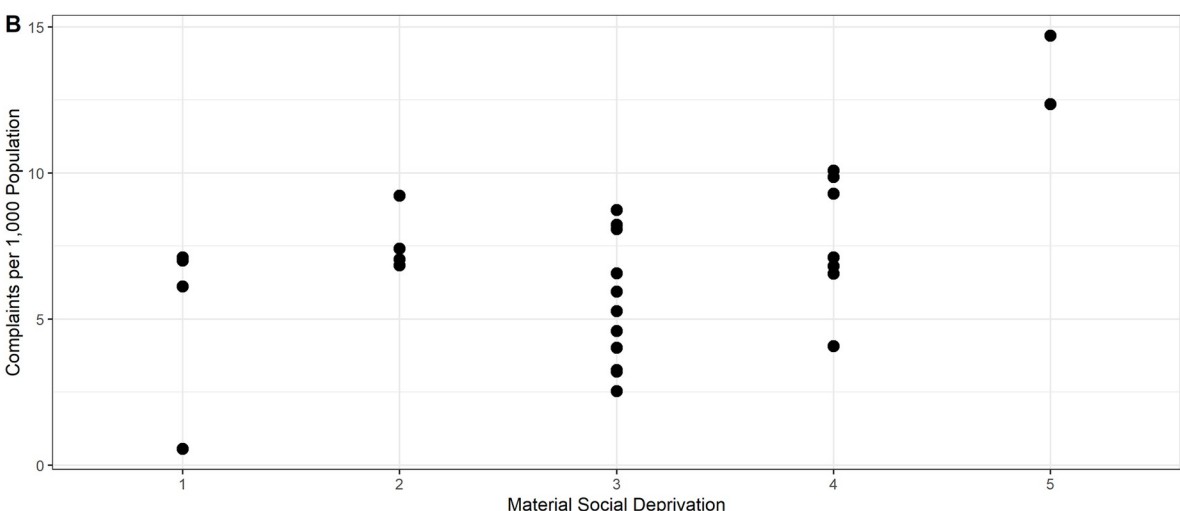

**Fig 3. Geographic distribution of complaints.** Panel (A) shows the number of complaints per 1,000 population by forward sortation area or FSA (first three digits of the postal code). FSAs are colored according to the level of deprivation and the size of each circle indicates the number of complaints per 1,000 population. Panel (B) shows a scatter plot of the FSA complaint rate versus deprivation. The background map (A) was sourced from OpenStreetMaps [19].

information (131 of 4285 complaints: 3.1%) about issues like how long landlords have to eliminate pests from a rental unit; (5) initiating a new service (111 of 4285 complaints: 2.6%), such as starting a public awareness program to educate transient residents (i.e., students, recent arrivals) about existing garbage programs; (6) requesting vegetation service (89 of 4285 complaints: 2.1%) most commonly cutting back overgrown vegetation and tree branches, and; (7) improving those vegetation services (7 of 4285 complaints: 0.2%). The categories 'other requests' (222 of 4285 complaints: 5.2%) and 'improving other services' (29 of 4285 complaints: 0.7%) included banning rodenticides and improving services that hold landlords accountable for the state of their rental properties. Complaints in these categories were comprised of requests that were not present in the random 20% of data the coders used to define the original coding framework for enumeration. Therefore, the content of these requests were not further categorized and were not counted because they arose after the qualitative coding framework was established. None of the four coders noted that requests to exterminate rats were significant enough of a concern to warrant its own distinct emergent theme.

When complainants requested actions, they suggested that either the city should be directly responsible for carrying out the action, or that they should enforce compliance or encourage a behaviour change by the offending individual. Actions that they felt the city should be responsible for included cleaning up alleyways, increasing the number of trash pickups, and maintaining vegetation on public property such as sidewalks. In comparison, many felt that the city should force compliance when there were issues regarding residential or rental properties. For example, complainants generally requested that the city force their neighbors or landlord to address existing rat problems and/or their associated causes including property maintenance issues, messy yards, and building deterioration/disrepair.

## 4. Discussion

In this study, we reviewed thousands of complaints made to the City of Vancouver to understand why and where people cared about rats and what they wanted the municipal government to do about them. Overall, we found that rats were a priority in a minority of cases. People were more concerned about broader community issues of which rats were one part. In general, complaints were made when problems were highly visible, nearby, and in and around living spaces. Complaints occurred across the entire city, but they were slightly higher rates in the most deprived neighborhoods, and they reflected residents' desire for the city to intervene to remediate specific problems related to rats.

### 4.1 Why did people complain about rats?

Rats were the primary reason for complaining when the perceived hazard was high. Complainants perceived this hazard to be highest when rats were a visible threat to the health, belongings, or property of themselves or their loved ones. The hazard of rats was considered to be especially important when rats were in or near complainants' residences or other locations where they spent time. However, complaints in which rats were the primary concern comprised a minority of cases. Instead, rats were viewed as symbolic of broader community issues that required action. In these complaints, rats were seen as only one symptom of a broader community problem and they were often used as a justification for why that broader problem was important. This is comparable to findings in London, UK where residents reported that while they found the presence of dog feces disgusting, they saw it as representative of wider neighborhood neglect by other residents and local authorities [20]. Although dog feces was a significant issue, the interviewees felt that the more important problem was the fact that people

didn't care about the state of their neighbourhood, one of the results of which was that people didn't clean up after their dogs.

Similarly, there is a growing body of evidence indicating that rats impact people within a variety of broader and intertwined community problems [10,12,21,22]. For example, German and Lakin [21] reported that rats were an indicator of environmental justice issues like neighborhood disorder and increased symptoms of depression. Likewise, interviewees in Byers et al. [10] felt that rats were a product of the disregard for their communities by the government which led to greater neighborhood deterioration. In the complaints reviewed in this study, people were primarily concerned about analogous issues linked to neighbourhood disorder, neglect, and deterioration such as garbage, waste, and compost problems, disregard for property maintenance, and overgrown vegetation. This often manifested as neighbors or landlords who did not maintain their properties and allowed them to become overgrown, cluttered, and dilapidated. In other circumstances, these neighborhood problems were seen to be caused by inefficient municipal services, ineffective trash pickup, or by people who were unhoused. Rats are connected to these broader community issues because they are able to thrive in places where such disorder provides a surplus of food, water, and harborage [23]. As such, although most people were primarily complaining about these broader community issues, they reported rats as one important symptom of the larger issue.

## 4.2 Rat-reduction objectives do not address residents' concerns

This study demonstrates that the common objective of current municipal rat management strategies to reduce rat populations at as large a scale as possible [12,24,25] is unlikely to address residents' actual concerns regarding rats. First, people only cared primarily about rats in specific instances where their hazard was perceived to be high, a fact which generalized rat reduction programs does not account for. For example, the predominant aim of these programs is to reduce rats in as many outdoor spaces as possible [22,26,27]. In contrast, this study found that rats were more likely to be a primary concern when they were in or around someone's living space, areas that may be the least likely to be addressed by these municipal programs [12,13]. This indicates that municipal rat management strategies would better serve their communities by targeting rat control efforts to specific instances in which the hazards posed by rats are perceived to be highest, including in and around places where people live. However, these hazards may be different in different cities and may change over time, such that ongoing consultation with residents may be required.

Second, rat management programs designed primarily with rat reduction objectives fail to address the wider community problems that residents were concerned about in this study. For example, most complainants were concerned with garbage problems, landlord neglect, neighborhood deterioration, and overgrown yards. This is important because municipal rat management programs have largely focused on extermination and have only addressed these wider community problems causing rats when they have surplus resources [12,13,24,28,29]. This prioritization of interventions, with rat extermination first and broader community issues second, is in the opposite direction of resident priorities.

Taken together, this suggests that municipal programs might better serve residents if designed with these broader community problems as the primary focus and with rat reduction in targeted cases. Not only would this more directly address the actual concerns of residents, but it would more effectively reduce rat problems by decreasing available sources of food, water, and harborage [23,28,30]. For instance, complaints were often made about neighbors who had messy, smelly, and overgrown yards that were also attracting rats. In this case, sending a pest management professional to kill the rats would not address the complainant's

primary concern. Further, because of rats' ability to move around cities, avoid lethal control measures, and their high birth rates, failing to address the unkempt and overgrown property would allow the rat population to quickly rebound [31,32]. Instead, seeking to address the messy yard and the reasons for its persistence would not only address the most important concern of the complainant, but it would also minimize the rat problem by removing their sources of food, water, and harborage. Targeted rat control efforts would still be important in those minority of cases where rats were residents' primary concern, because they were, for example, in their home.

Addressing residents' concerns about broader community issues would have many other co-benefits for residents beyond simply reducing rat populations. A randomized control trial in Philadelphia, Pennsylvania found that cleaning up vacant lots (i.e., removing garbage, grating soils, planting new vegetation, fencing the perimeter), and regularly maintaining them, resulted in those living nearby reporting lower levels of depressive symptoms and better overall mental health compared to people living near vacant lots that received no intervention [33]. Initiatives like this, designed to give people a cleaner and healthier community, would have a variety of such co-benefits and would also reduce the size of the rat population.

## 4.3 Locations of complaints

The highest and lowest rates of complaints were observed in areas with the highest and lowest levels of neighborhood deprivation, respectively, consistent with previous literature [34,35]. This is unsurprising because the most deprived areas often have higher levels of food, water, and harborage which allow rats to thrive [23,36,37]. However, rat complaints arose in areas with all levels of deprivation, indicating that rats were problematic in all areas of the city. The finding that the rate of complaints was the lowest and highest in the most privileged and deprived neighborhoods, respectively, may also reflect the ability of residents living in these areas to address problems on their own. People living in the most deprived areas may have the least resources to address problems on their own (e.g., pay for pest control services) and may therefore be more likely to rely on city services to address rat problems.

Within these neighborhoods, the number of complaints about owned and rented residential properties were greater than the number of complaints about streets, sidewalks, and alleyways. This is important because many municipal rat management programs have aimed to reduce rats outside in shared spaces like alleyways and streets, with less resources towards addressing issues on private properties including inside and around peoples' living spaces [12]. Further, residents who complained about rats on streets, sidewalks, and alleyways were more concerned with broader community issues, while residents who complained about rats in relation to their home were more likely to cite concerns about rat-associated hazards. This underscores a growing awareness that not all rat-human interfaces–situations in which rats and people interact– present the same threats or risks. For instance, Himsworth [38] describes how in most cases, rats are a relatively minor threat to people, becoming problematic in certain situations such as when inside the home. Our findings align with this view, as residents were more concerned with rats in their homes than they were with rats outdoors. Another example in which rats might pose a minor threat to people is in sewers. While these rats are relatively understudied [39], sewers were never mentioned in these complaints, indicating that there may not be an important interface between sewer rats and people in Vancouver. As such, sewer rat control programs that exist in other areas [24] may not be worth prioritizing in this city. Instead, these complaints suggest that rat management programs will better serve residents' needs by only providing rat control services in specific instances when they are in peoples' homes. While it might be outside of the jurisdiction of a program to send a pest management professional into someone's house, municipalities do have responsibilities to design and enforce bylaws that

hold residents and landlords accountable to state of their property and its impacts on tenants and neighbors [13].

## 4.4 Actions requested

Complaints were made when residents wanted the municipal government to intervene to fix a broad community issue. This aligns with the finding that the highest rates of complaints came from the most deprived areas where people may have the least resources to address rat problems on their own and where these community issues may be more common. Accordingly, complainants either requested that the municipality fix the issues directly (i.e., send people to clean up the garbage issue, cut back overgrown vegetation) or that they force the responsible resident to address the issue (i.e., force their neighbor to clean their yard, force their landlord to improve their property). This highlights the importance of expanding management efforts beyond traditional lethal techniques to refocus on other activities. Many previous authors have recognized the need to develop and enforce bylaws about rats and associated issues in order to effectively manage urban rat problems [13,22,40,41]. In New York City, management efforts are centered on locating environmental conditions conducive to rat populations and forcing property owners to fix them [42]. In instances where the municipality is directly responsible for services (i.e., garbage pickup, waste bin supply and maintenance, park and vegetation management), rat management programs might serve to identify gaps in those services contributing to the rat population and alert the responsible municipal department of the need for improvements. Such a function would have the co-benefit of serving as an important evaluation metric for many different municipal services that rats affect.

## 4.5 Community consultation

This study emphasizes the importance of human dimensions in municipal rat management and the need to focus management objectives on the actual needs and desires of residents. With current rat reduction objectives, municipal rat programs are often underfunded relative to their lofty mandates to reduce rats across entire cities [13,26,41]. This study suggests that prioritizing rats is a challenge for communities that face many other and often more pressing issues. As such, management programs focused on reducing rats are likely to become a low priority for residents living in these neighborhoods. This may be a key reason that municipal rat programs are so often underfunded [13,26,28,41]–they are not designed to address the issues the urban residents are most concerned about. Municipalities who consult with communities will be best positioned to address their needs and to create approaches that manage the suite of intersecting issues that residents prioritize.

Community consultation has been helpful in the management of other urban wildlife species. For example, residents of Banff, Canada formed a community-based advisory committee to work with Parks Canada to manage growing elk populations which were causing significant ecological and public safety problems [43]. Community consultation not only helped to increase community awareness of the problem, but it also helped to prioritize areas where elk populations should be reduced because of high elk-human conflict and actions to take to reduce that conflict. While this work is not directly analogous to urban rat problems, the community-based advisory committee worked to develop strategies specifically to reduce the significance of elk-human interactions resulting in negative conflict. In the context of this study, we found that rat management objectives would be better framed around addressing broader community health problems and providing more direct rat management services in instances where rats pose an immediate hazard to residents because of close contact, similar to reducing elk-human conflict in Banff. However, it isn't clear how the content of the rat complaints

reviewed in this study may vary across space and time. Future studies should examine community needs with regard to rats in other cities and explore how those needs can be integrated into municipal management programs.

### 4.6 Limitations

This study had important limitations. First, we were unable to determine the degree to which these complaints were representative of rat issues experienced by the general population in Vancouver. Many people in the city might not know about these complaint services and the people who complain may differ in values or characteristics from those who don't. For example, the association between complaints and increasing deprivation may reflect the fact that people living in privileged areas address rat problems on their own instead of reporting complaints to the city. Future work should focus on getting a more representative sample of the population by providing phone operators with a standardized set of questions, deploying a questionnaire to a randomly selected sample of the population, or implementing a dedicated rat/wildlife hotline. Second, most complainants (69.6%) did not share a desired action, and we could not determine the exact motivations for the complaints. However, it seems unlikely that residents were making complaints to the city unless they wanted the city to take action to affect change over the problem they were complaining about. Further, complaints made by phone were summarised by a 311 operator, who then referred the complaint to the responsible municipal department. It may be the case that the operator did not record the action desired and instead relied on the responsible department to determine what actions to take. Third, because we included only secondary use of complaint data, we inferred the primary reasons for complaining based on what was written. Municipalities looking to develop their own rat management strategies with objectives that align with their citizens' needs would benefit from directly engaging with residents in the design of their programs.

## 5. Conclusions

This study shows that urban rats are integrated with residents' concerns about many other aspects of urban life and municipal services. We demonstrate the utility of using municipal complaint data to better understand the needs and desires of urban residents and we suggest that this information be used by cities to develop municipal management programs which more directly manage the issues that people care about. Overall, we present evidence that municipal rat management programs designed primarily to reduce rats across cities in as many outdoor spaces as possible are unlikely to meet the varied needs and desires of residents. Existing programs designed with this objective may face inevitable resource limitations and funding challenges due to low prioritization. The complaints reviewed in this study suggest the need for programs to expand management efforts beyond traditional extermination methods because they may not address residents' concerns. Instead, to meet citizen needs more directly, cities should refocus on activities that address broader community problems including bylaw development, bylaw enforcement, and ongoing evaluation and improvement of municipal services associated with rat issues. Direct rat control services might best be directed to specific instances in which the hazards associated with rats are highest, such as when they are in and around peoples' livings spaces and properties.

## Supporting information

**S1 Table. Complaint categories pre-defined by the City of Vancouver.** The City used these categories to categorize complaints according to their contents.
(DOCX)

## Author Contributions

**Conceptualization:** Michael Joseph Lee, Kaylee A. Byers, Susan M. Cox, Chelsea G. Himsworth.

**Data curation:** Michael Joseph Lee, Lisa K. F. Lee.

**Formal analysis:** Michael Joseph Lee, Kaylee A. Byers, Xiaocong Guo, Lisa K. F. Lee.

**Funding acquisition:** Michael Joseph Lee, Chelsea G. Himsworth.

**Investigation:** Michael Joseph Lee.

**Methodology:** Michael Joseph Lee, Xiaocong Guo, Lisa K. F. Lee, Susan M. Cox, Chelsea G. Himsworth.

**Project administration:** Michael Joseph Lee.

**Supervision:** Chelsea G. Himsworth.

**Visualization:** Michael Joseph Lee.

**Writing – original draft:** Michael Joseph Lee.

**Writing – review & editing:** Michael Joseph Lee, Kaylee A. Byers, Chelsea G. Himsworth.

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
