## [Decision Letter · Decision Letter 0]

18 Sep 2023

PONE-D-23-21231Urban rats are the ‘fall-guy’: Resident motivations for municipal rat complaintsPLOS ONE

Dear Dr. Lee,

Thank you for submitting your manuscript to PLOS ONE. After careful consideration, we feel that it has merit but does not fully meet PLOS ONE’s publication criteria as it currently stands. Therefore, we invite you to submit a revised version of the manuscript that addresses the points raised during the review process.

Authors need to explain the methodology and data more carefully for a general reader's understanding. Furthermore, the academic novelty needs to be highlighted. 

We look forward to receiving your revised manuscript.

Kind regards,

Muhammad Khalid Bashir, PhD

Academic Editor

PLOS ONE

Journal Requirements:

“This work was funded by the City of Vancouver’s 2018-2020 Street Cleaning Grant Program in support of their efforts to establish suitable rat management protocols for Vancouver.  “

“This work was funded by the City of Vancouver’s 2018-2020 Street Cleaning Grant Program in support of their efforts to establish suitable rat management protocols for Vancouver. The funders had no role in study design, data collection and analysis, decision to publish, or preparation of the manuscript.”

4. We note that Figure 3 in your submission contain [map/satellite] images which may be copyrighted. All PLOS content is published under the Creative Commons Attribution License (CC BY 4.0), which means that the manuscript, images, and Supporting Information files will be freely available online, and any third party is permitted to access, download, copy, distribute, and use these materials in any way, even commercially, with proper attribution. For these reasons, we cannot publish previously copyrighted maps or satellite images created using proprietary data, such as Google software (Google Maps, Street View, and Earth). For more information, see our copyright guidelines: http://journals.plos.org/plosone/s/licenses-and-copyright.

a. You may seek permission from the original copyright holder of Figure 3 to publish the content specifically under the CC BY 4.0 license. 

Reviewers' comments:

Reviewer's Responses to Questions

**Comments to the Author**

1. Is the manuscript technically sound, and do the data support the conclusions?

Reviewer #1: Yes

Reviewer #2: No

2. Has the statistical analysis been performed appropriately and rigorously? 

Reviewer #1: N/A

Reviewer #2: No

3. Have the authors made all data underlying the findings in their manuscript fully available?

Reviewer #1: Yes

Reviewer #2: No

4. Is the manuscript presented in an intelligible fashion and written in standard English?

Reviewer #1: Yes

Reviewer #2: Yes

5. Review Comments to the Author

Reviewer #1: This research has encompass the issue of rats very well. Its method are appropriate and qualitative procedures are well adopted. The researchers created a combined index of material and social deprivation using the first strategy recommended in the MSDI user guide but in results section, findings related to this index are not highlighted.

The manuscript is well written.

Reviewer #2: The authors have attempted to study an interesting issue. However, the design of the study and the nature of data don't meet the standards of sufficient academic rigor. Data analysis is limited to summary statistics.

6. PLOS authors have the option to publish the peer review history of their article (what does this mean?). If published, this will include your full peer review and any attached files.

Reviewer #1: **Yes: **Dr. Saima Afzal

Reviewer #2: No

---

## [Author Response · Author response to Decision Letter 0]

12 Oct 2023

Editor Comments:

Authors need to explain the methodology and data more carefully for a general reader's understanding. Furthermore, the academic novelty needs to be highlighted. 

Author Response: Thank you for taking the time to review our manuscript. We have addressed this helpful feedback throughout the manuscript, with responses below. First, we have revised the Methods section to more clearly explain the methodology and data for general readers. We have also moved the supplementary table describing the qualitative coding process into the main text so that the methods are clear without needing to go to the supporting information. Second, we have revised and shortened the Introduction section to clarify why this study is important and to highlight its academic novelty. 

“This work was funded by the City of Vancouver’s 2018-2020 Street Cleaning Grant Program in support of their efforts to establish suitable rat management protocols for Vancouver. “

“This work was funded by the City of Vancouver’s 2018-2020 Street Cleaning Grant Program in support of their efforts to establish suitable rat management protocols for Vancouver. The funders had no role in study design, data collection and analysis, decision to publish, or preparation of the manuscript.”

Author Response: I have removed the funding information from the acknowledgements. Please keep the funding statement as is stated immediately above this response. 

Author Response: I have included captions for the supporting information at the end of the manuscript.

4. We note that Figure 3 in your submission contain [map/satellite] images which may be copyrighted. All PLOS content is published under the Creative Commons Attribution License (CC BY 4.0), which means that the manuscript, images, and Supporting Information files will be freely available online, and any third party is permitted to access, download, copy, distribute, and use these materials in any way, even commercially, with proper attribution. For these reasons, we cannot publish previously copyrighted maps or satellite images created using proprietary data, such as Google software (Google Maps, Street View, and Earth). For more information, see our copyright guidelines: http://journals.plos.org/plosone/s/licenses-and-copyright.

a. You may seek permission from the original copyright holder of Figure 3 to publish the content specifically under the CC BY 4.0 license. 

Author Response: The background map used in Figure 3 is sourced from ‘OpenStreetMaps’ (Hyperlink). These maps are open data and are freely available for use under the Creative Commons Attribution-ShareAlike 2.0 license (CC BY-SA 2.0). I have included a citation for OpenStreetMaps in the revised manuscript in the figure 3 caption. This data is “free to copy, distribute, transmit and adapt … as long as you credit OpenStreetMap and its contributors.” 

Reviewer 1:

1. Is the manuscript technically sound, and do the data support the conclusions?

Reviewer #1: Yes

2. Has the statistical analysis been performed appropriately and rigorously? 

Reviewer #1: N/A

3. Have the authors made all data underlying the findings in their manuscript fully available?

Reviewer #1: Yes

4. Is the manuscript presented in an intelligible fashion and written in standard English?

Reviewer #1: Yes

5. Review Comments to the Author

Reviewer #1: This research has encompass the issue of rats very well. Its method are appropriate and qualitative procedures are well adopted. The researchers created a combined index of material and social deprivation using the first strategy recommended in the MSDI user guide but in results section, findings related to this index are not highlighted.

The manuscript is well written.

Author Response: Thank you for taking the time to review our manuscript. To address your main concern, we have further explained in the Methods section how we assessed the deprivation data. We have also more clearly highlighted the results related to the deprivation index in the Results section and in the Discussion. Overall, we found that there was some evidence that complaints were more common in areas with higher levels of deprivation. This is consistent with expectations based on the previous literature and we discuss the implications of this in section 4.3 in the Discussion. 

Reviewer 2:

1. Is the manuscript technically sound, and do the data support the conclusions?

Reviewer #2: No

2. Has the statistical analysis been performed appropriately and rigorously? 

Reviewer #2: No

3. Have the authors made all data underlying the findings in their manuscript fully available?

Reviewer #2: No

4. Is the manuscript presented in an intelligible fashion and written in standard English?

Reviewer #2: Yes

5. Review Comments to the Author

Reviewer #2: The authors have attempted to study an interesting issue. However, the design of the study and the nature of data don't meet the standards of sufficient academic rigor. Data analysis is limited to summary statistics.

Author Response: Thank you for taking the time to review our manuscript. To address your concerns about the nature of these data, the design of the study, and the use of summary statistics we have further clarified in the manuscript that this is a qualitative study in which the primary objective of the analysis was to describe the content of these textual complaints made to the City about rats. We use a method called ‘qualitative description’ which we support with references in the Methods section. This method focuses on summarizing the breadth of content in these complaints to understand why different people in the City of Vancouver cared enough about rats to complain about them. As a result, we only provide summary statistics so that readers can get a sense of the breadth of the different kinds of complaints. We do not perform further statistical analysis because these complaints were not collected from a representative sample of the general population and thus statistical conclusions could lead to erroneous inferences about the distribution of these rat problems in the wider population.

---

## [Decision Letter · Decision Letter 1]

26 Dec 2023

Urban rats are the ‘fall-guy’: Resident motivations for municipal rat complaints

PONE-D-23-21231R1

Dear Dr. Lee,

We’re pleased to inform you that your manuscript has been judged scientifically suitable for publication and will be formally accepted for publication once it meets all outstanding technical requirements.

Kind regards,

Muhammad Khalid Bashir, PhD

Academic Editor

PLOS ONE

Additional Editor Comments (optional):

Reviewers' comments:

Reviewer's Responses to Questions

**Comments to the Author**

1. If the authors have adequately addressed your comments raised in a previous round of review and you feel that this manuscript is now acceptable for publication, you may indicate that here to bypass the “Comments to the Author” section, enter your conflict of interest statement in the “Confidential to Editor” section, and submit your "Accept" recommendation.

Reviewer #3: All comments have been addressed

2. Is the manuscript technically sound, and do the data support the conclusions?

Reviewer #3: Partly

3. Has the statistical analysis been performed appropriately and rigorously? 

Reviewer #3: Yes

4. Have the authors made all data underlying the findings in their manuscript fully available?

Reviewer #3: Yes

5. Is the manuscript presented in an intelligible fashion and written in standard English?

Reviewer #3: Yes

6. Review Comments to the Author

Reviewer #3: I have looked at the reviewer comments and compared original version with the revised one. I am satisfied with the revision and paper can be accepted.

7. PLOS authors have the option to publish the peer review history of their article (what does this mean?). If published, this will include your full peer review and any attached files.

Reviewer #3: No

---

## [Editor Report · Acceptance letter]

9 Jan 2024

PONE-D-23-21231R1 

PLOS ONE

Dear Dr. Lee, 

I'm pleased to inform you that your manuscript has been deemed suitable for publication in PLOS ONE. Congratulations! Your manuscript is now being handed over to our production team.

Kind regards, 

on behalf of

Dr. Muhammad Khalid Bashir 

Academic Editor

PLOS ONE